# Impact of Relational Coordination on Job Satisfaction and Willingness to Stay: A Cross-Sectional Survey of Healthcare Professionals in South Tyrol, Italy

**DOI:** 10.3390/bs14050397

**Published:** 2024-05-10

**Authors:** Christian J. Wiedermann, Verena Barbieri, Adolf Engl, Giuliano Piccoliori

**Affiliations:** 1Institute of General Practice and Public Health, Claudiana—College of Health Professions, 39100 Bolzano, Italy; 2Department of Public Health, Medical Decision Making and Health Technology Assessment, University of Health Sciences, Medical Informatics and Technology, 6060 Hall, Austria

**Keywords:** relationship coordination, job satisfaction, workforce retention, health personnel, interprofessional relations, organizational culture

## Abstract

Job satisfaction and willingness to stay are critical for workforce stability in a challenging healthcare environment. This study examined how relational coordination, a key factor in teamwork and communication, influences outcomes among healthcare professionals in a bilingual, culturally mixed region of Italy. This cross-sectional survey included general practitioners, hospital physicians, nurses, and administrators from the South Tyrol Health Service, using the ‘Relational Coordination Survey’ and additional measures of job satisfaction and willingness to stay. The analytical methods used included descriptive statistics, correlations, and regression analyses. This study applied path analysis, including mediation and moderation techniques, to investigate the roles of relational coordination and job satisfaction in influencing the willingness to stay. It employs Conditional Process Analysis with the PROCESS macro in SPSS, focusing on models for moderated mediation analysis. The results indicated a critical influence of relational coordination on both job satisfaction and willingness to stay among the 525 healthcare professionals. Job satisfaction varied by health district and years of service, with midcareer professionals being the least satisfied. The findings highlight the central role of relational coordination in job satisfaction and willingness to stay and confirm that low job satisfaction increases turnover intentions. Relational coordination directly enhanced job satisfaction and willingness to stay, while also serving as a mediating factor that amplifies the impact of job satisfaction on retention intentions. This study reinforces the need for strong teamwork and communication to stabilize the healthcare workforce. Targeted interventions aimed at improving relational coordination could significantly enhance job satisfaction and retention among healthcare professionals, particularly in culturally diverse settings such as South Tyrol.

## 1. Introduction

In the complex field of healthcare, communication and coordination are essential for patient care and system efficiency [1]. As the complexity of healthcare increases, improving relational coordination (RC) has become critical. Research underscores the importance of examining RC not only as an organizational asset, but also as a potential mitigator of global problems such as violence against healthcare workers [2].

RC is a framework that theorizes the importance of relationships in coordinating complex tasks and includes key dimensions such as frequent, accurate communication and shared goals that are essential in healthcare, where teamwork is critical to patient safety and quality of care. Studies show that higher levels of RC correlate with improved care and patient satisfaction [3].

The role of RC in supporting a collaborative healthcare environment is well documented, with benefits including improved work outcomes, teamwork, quality of patient care, staff work experience, reduced turnover, and increased job satisfaction [4].

Understanding the dynamics within and between healthcare teams is critical. Intra-group RC refers to coordination between members of the same team, whereas inter-group RC involves coordination between different teams or departments, each of which affects work outcomes such as satisfaction and retention differently [4].

The South Tyrolean Health Agency (SABES-ASAA) faces specific challenges that affect RC, job satisfaction, and retention, such as language barriers, resource constraints, and rapid technological change [5]. This study aims to explore the relationship between RC, job satisfaction, and retention in the unique context of South Tyrol, focusing on how RC dynamics documented in a recent cross-sectional survey influence these key elements [6].

Understanding the dynamics within and between healthcare teams is important for improving job satisfaction and overall organizational performance. Within-group RC refers to the depth of coordination among members of the same team, focusing on shared goals, mutual respect, and frequent, timely, and accurate communication that enhances internal group functioning. Conversely, intergroup RC extends these principles across different teams or departments, emphasizing the importance of intergroup cooperation to achieve broader organizational goals. Research suggests that while both types of coordination are essential, their effects on work outcomes such as satisfaction and retention may differ significantly due to their different scope and interaction dynamics [4,7].

This analysis will examine whether demographic and job-related factors, such as age, gender, professional role, and health district, influence the relationships between RC, job satisfaction, and retention intentions. By understanding these dynamics, this study aims to provide insights for improving team dynamics in South Tyrol and potentially in similar healthcare settings worldwide. To further elucidate the dynamics of RC, job satisfaction, and retention among healthcare professionals, this study also examines whether demographic and job-related factors moderate the relationships among these variables. The question is whether factors such as age, gender, professional role, and health district may influence the strength and direction of the mediation effects observed between RC, job satisfaction, and retention intentions. By testing these hypotheses, this study aims to provide meaningful insights into scholarly conversation on RC in healthcare settings and offer practical recommendations to improve team dynamics in South Tyrol and beyond.

## 2. Methods

Building on the recently presented research [6], this study extends the exploration of RC among healthcare professionals within the SABES–ASAA. The first study assessed the status of RC among physicians, nurses, and administrators, laying the groundwork for understanding the broader impact of RC in diverse multilingual healthcare environments. It detailed the methodological approach, participant demographics, and intricacies of administering the RC survey in this unique setting [8]. The survey included items on job satisfaction and intention to stay, which were collected, but were not the focus of the analysis. The second paper focuses on these areas, with the aim of elucidating how RC among healthcare professionals correlates with their job satisfaction and intention to continue working within the SABES–ASAA.

### 2.1. Study Design and Participants

This analysis completes the investigation within the same research context using the results of a cross-sectional observational design to explore the dynamics of the healthcare system in greater depth.

Participating health professionals who are actively involved in healthcare delivery within the SABES–ASAA include GPs, HPs, nurses, and administrative personnel who are directly involved in outpatient patient care. The selection criteria excluded non-clinical staff who were not directly involved in patient care, and professionals working outside the public health system. The recruitment process included targeted e-mail invitations from the four health districts of South Tyrol. Employed specialists, nurses, and administrative staff were invited by the SABES–ASAA, whereas GPs contracted by the health authority received invitations from the Institute for General Practice and Public Health. To increase participation, reminders were sent after the initial invitation. The response rate for the survey was 26%, with a total of 525 completed online responses, as described previously [8].

### 2.2. Questionnaire

The questionnaire used was the Relational Coordination Survey (RCS), a validated instrument that assesses seven dimensions of communication and relationships integral to RC within healthcare teams (Table 1). Its findings on frequency, timeliness, accuracy of communication, problem solving, shared goals, shared knowledge, and mutual respect among healthcare professionals in South Tyrol provide the context for this second analysis [8].

Specific questions were included in the survey to measure the participants’ job satisfaction and intent to stay. One item asked, ‘Overall, how satisfied are you with your current position?’ with responses on a 5-point scale: (a) very dissatisfied, (b) dissatisfied, (c) moderately satisfied, (d) fairly satisfied, and (e) completely satisfied. The other item asked about their future plans with the statement, ‘I plan to leave the South Tyrolean Health Service/contract with the region as soon as possible,’ with responses on a 5-point scale ranging from (a) strongly agree to (e) strongly disagree. These additional questions complement the results of the RCS and provide a more complete understanding of how RC affects not only teamwork and communication, but also the overall job satisfaction and future commitment of health professionals within the South Tyrol Health Service. Because all questions on RC, job satisfaction, and intention to stay were obligatory, information loss was minimized.

### 2.3. Data Analysis

Data on RC, job satisfaction, and intention to stay in the SABES–ASAA were analyzed using descriptive statistics to provide an overview of the assessments. For descriptive statistics, job satisfaction and intention to stay were dichotomized to provide binary results. For job satisfaction, responses indicating “very satisfied” and “completely satisfied” were categorized as “satisfied” (1), while “very dissatisfied”, “dissatisfied”, and “average satisfaction” were categorized as “not satisfied” (0). For willingness to stay, responses of ‘strongly agree’ and ‘somewhat agree’ to continue working with the current employer were classified as ‘willing to stay’ (1), while ‘strongly disagree’, ‘somewhat disagree’ and ‘neither agree nor disagree’ were classified as ‘undecided to stay’ (0). Chi-square tests were used to compare dichotomous groups, and the Wilcoxon rank sum test and Mann–Whitney test were used to compare metric variables between dichotomous groups.

To understand the relationships between RC and RC subscales and the two main outcomes, job satisfaction and intention to stay, correlational analyses using either Pearson’s or Spearman’s methods were used to assess the linear relationships and provide correlation coefficients.

Linear regression analyses were conducted to determine the extent to which RC and demographic or job-related factors could explain the variance in job satisfaction and intention to stay.

Path analysis was used to further explore these relationships [9]. Mediation [10] and moderation [11] analyses were conducted to examine the potential mediating role of job satisfaction in the relationship between RC and retention, with RC as a mediator of the effect of job satisfaction on retention. Both mediation and moderation analyses were conducted to explore the dynamics between RC, job satisfaction, and retention. Specifically, the potential of job satisfaction to mediate the relationship between RC and retention and, in a separate model, the possibility of RC acting as a mediator in the relationship between job satisfaction and retention were explored. Furthermore, the models were extended to identify any demographic or job-related factors that might moderate these relationships. The analyses employed Conditional Process Analysis (CPA) using the PROCESS macro in SPSS developed by Hayes [12]. Specifically, we used models 1, 4, 7, 8, 14, and 15 of PROCESS, which allowed for the examination of a moderated mediation framework differentiating across different job roles or language groups. The effects were assessed for statistical significance using bootstrapping with 5,000 resamples to generate confidence intervals for indirect effects, as recommended for robustness to normality violations.

All statistical analyses were performed using the IBM SPSS Statistics for Windows (version 25.0; IBM Corp., Armonk, NY, USA).

## 3. Results

### 3.1. Descriptive and Work-Related Overview

Table 2 presents an overview of the descriptive results of the investigation of the relationships among RC, job satisfaction, and retention among healthcare professionals in the South Tyrolean healthcare system. The analyses conducted on the responses of GPs, HPs, administrative staff, and nurses provided a breakdown of job satisfaction and willingness to stay across different professional, linguistic, and demographic segments as well as their association with different aspects of RC.

Dichotomized job satisfaction was significantly different between health districts and subgroups of years of service, with the middle (10–20 years in service) being the most unsatisfactory group, whereas there was no significant difference between professional areas, languages, and gender. Dichotomized willingness to stay did not show any significant differences for demographic factors.

Between the two categories of the dichotomized variable job satisfaction, ‘satisfied’ and ‘not satisfied’, highly significant differences were found for RC, within-group RC, and between-group RC, as well as for the RC dimensions ‘accuracy of information’, ‘timeliness of information’, ‘problem-solving communication’, ‘shared goals’, ‘shared knowledge’, and ‘mutual respect’. Only the dimension ‘frequency of information’ produced fewer but still significant results (Figure 1).

For the dichotomized variable willingness to stay, ‘willing to stay’ and ‘undecided to stay’, RC and between-group RC showed a highly significant difference between the two categories, while no significant difference was found for within-group RC (Figure 2). The differences were found in the RC dimensions ‘accuracy of information’, timeliness of information’, ‘problem-solving communication’, ‘shared goals’, ‘shared knowledge’, and ‘mutual respect’. Only the dimension ‘frequency of information’ produced fewer but still significant differences.

### 3.2. Correlations between Relatinal Coordination, Job Satisfaction and Willingness to Stay

Correlational analyses of the relationships between RC and the outcomes of job satisfaction and intention to stay are shown in 32 to highlight the importance of RC in shaping the work environment and the overall satisfaction and loyalty of healthcare professionals within the South Tyrol Health Service. Job satisfaction and willingness to stay were significantly correlated (Spearman’s ρ = 0.654, *p* < 0.001; Pearson’s r = 0.638, *p* < 0.001).

To analyze the predictive power of RC on job satisfaction and intention to stay, according to the correlations in Table 3, simple linear regression analyses were used to quantify the extent to which different RC dimensions explained the variance in the scores. The results are presented in Table 4 and Table 5, respectively.

Willingness to stay and RC scores, including overall, within-group, and between-group coordination, significantly predict job satisfaction, with willingness to stay being a strong predictor (B = 0.518, *p* < 0.001) (Table 4). Within-group RC showed a lower impact (B = 0.335) than between-group RC (B = 0.734), suggesting that how individuals perceive their immediate team’s coordination can affect their job satisfaction differently. Each RC dimension—frequency, timeliness, accuracy of communication, problem-solving, shared goals, knowledge, and mutual respect—also significantly contributes to job satisfaction, highlighting the multifaceted nature of the impact of RC. The Durbin–Watson statistic indicates acceptable autocorrelation levels in the residuals for each predictor.

Job satisfaction is a significant predictor of intention to stay, with a strong effect size (B = 0.784), indicating that higher job satisfaction is associated with greater intention to stay. While overall RC and between-group RC were marked as significantly predictive, within-group RC was not a significant predictor. All specific dimensions of RC, such as timeliness and accuracy of communication, problem-solving communication, shared goals, and mutual respect, significantly predicted the intention to stay.

### 3.3. Models of Linear Regression, Moderation, and Mediation for Willingness to Stay

Willingness to stay as a dependent variable (Y) was analyzed with linear regression as well as path analysis to understand how it is affected by RC and job satisfaction. Furthermore, the variable ‘health district’ was split into four dummy variables (health district–0, health district–1, health district–2, and health district–3), and its impact was investigated in the different models. Health district–0 (Kendall’s tau b = 0.09, *p* < 0.05) and health district–2 (b = 0.12, *p* < 0.01) were significantly correlated with job satisfaction, but none of the four districts were correlated with willingness to stay. Furthermore, health district–0 (b = −0.077, *p* < 0.05), health district–1 (b = 0.094, *p* < 0.01), and health district–2 (b = 0.094, *p* < 0.05) were significantly correlated with RC. Age, gender, professional area, language, and years of service were not correlated with willingness to stay or job satisfaction; thus, they were not included in the analyses.

After the regression analysis, path analysis was conducted to further explore the relationships between RC, job satisfaction, and willingness to stay as dependent variables (Y). The applied path models are illustrated in Figure 3.

Table 6 presents the results of the linear regression and path analysis. The models explored how job satisfaction might affect the relationship between RC and willingness to stay and vice versa, and how RC might affect the relationship between job satisfaction and willingness to stay through mediation and moderation analyses.

Linear regression analysis, using RC and job satisfaction as predictors of willingness to stay, yielded an R² value of 0.42, indicating a moderate fit. However, in the moderation analysis of Model 1, which corresponds to the former linear regression model using job satisfaction and RC as independent predictors, amplification with the interaction term between neither RC (X) nor the interaction term between RC and job satisfaction (X) showed significant predictive power. This suggests that the simple linear model’s predictive capability did not extend to the moderation model in which interaction effects were considered.

In Model 4 of the mediation analysis, RC acted as a mediator between job satisfaction and willingness to stay. This model accounted for a substantial portion of the variance with an R² of 0.406, indicating a strong mediating effect of RC on the relationship between job satisfaction and willingness to stay. Conversely, when examining job satisfaction as a mediator in the RC-to-willingness-to-stay relationship, the model showed less explanatory power, with an R² of 0.137. This difference in R² values between the models suggests that it makes sense to continue with extended models using job satisfaction as a predictor and RC as a mediator when predicting employee willingness to stay.

In mediation Model 4 (N = 518), when ‘health district’ (W) was included as a covariate, job satisfaction (X) was used as a predictor, and RC (M) was included as a mediator to predict willingness to stay (Y), the influence of the health district was found to be non-significant. This suggests that health district, as an independent factor, does not have a statistically significant impact on healthcare professionals’ willingness to remain in their position when controlling for job satisfaction with RC as a mediator.

Model 7 is an extension of Model 4 including ‘health districts’ as dummy mediators (with Health District–0 as the baseline) on the path from predictor job satisfaction (X) to mediator RC (M). The model achieved an R² value of 0.425. However, health district as a moderator exhibits only a significant effect of health district–1 on RC and an interaction effect with job satisfaction for health district–1. The direct path from job satisfaction to willingness to stay is significant, indicating a strong direct relationship. In addition to the already known effect of the mediator RC, the small indirect effect of health district–1 suggests that health districts can have a nuanced role in shaping the influence of moderated job satisfaction on retention.

Model 14 extended Model 4, including health district as a moderator on the path from moderator RC to the dependent variable retention, revealing nuanced relationships. The model demonstrates that, while job satisfaction directly influences willingness to stay, the effect of RC alone on willingness to stay is not significant. However, health district–2 and 3 significantly impacted willingness to stay negatively, indicating regional differences. The interactions between RC and health districts further indicate that district-specific factors can significantly modify the relationship between RC and the willingness to stay.

Models 8 and 15 (both N = 480) explore the moderating effect of health districts (W) on the relationship between job satisfaction (X), RC (M), and willingness to stay (Y). Both models specifically examine how health districts influence both the direct paths from job satisfaction to willingness to stay and the mediated paths from job satisfaction to RC and RC to willingness to stay, respectively. However, the analysis revealed that the moderating effect of health districts on the direct pathway was not significant, suggesting that the impact of job satisfaction on willingness to stay was consistent across different health districts.

## 4. Discussion

This study revealed significant relationships among RC, job satisfaction, and willingness to stay among healthcare professionals in South Tyrol, demonstrating the central role of job satisfaction. Notable findings include the variance in job satisfaction across health districts and years of service, with midcareer professionals being the least satisfied. RC has a significant impact on job satisfaction and willingness to stay, with different dimensions affecting these outcomes. The findings clarify that health districts do not directly influence the relationship between job satisfaction and retention among health professionals. Instead, the effect of health districts is mediated by differences in RC across districts. This indirect influence suggests that while job satisfaction is a critical factor in retention, the effectiveness of RC, which varies across districts, plays a complementary role. Specifically, the way in which RC functions within each health district shapes the overall retention dynamic, and provides implications for understanding and addressing retention factors in the South Tyrolean healthcare system.

### 4.1. Job Satisfaction and Relational Coordination

In light of the findings of Ran et al. [13], job satisfaction played a mediating role between burnout and turnover intention among primary healthcare workers. Similar studies across diverse global regions have shown that enhancing RC not only boosts job satisfaction and reduces turnover intentions but also improves patient care quality. In the traditional view, RC acts as a mediator between organizational structures and outcomes. In the present study, however, RC was examined as a mediator directly influencing job satisfaction and retention among health professionals in the unique context of the different health districts of South Tyrol. These districts differ not only in organizational structures, but also in cultural and linguistic composition, with more German-speaking districts serving peripheral and rural areas and more Italian-speaking districts covering urban areas with larger hospitals and centralized services. This diversity adds complexity to RC dynamics and potentially influences the effectiveness of cross-cutting versus siloed structures in supporting or weakening RC.

Linguistic differences between predominantly Italian and predominantly German-speaking districts significantly influence RC [6]. German-speaking districts showed stronger RC, probably due to greater linguistic homogeneity, which facilitates more effective communication within healthcare teams. The study showed a clear effect of language on between-group RC, suggesting that shared linguistic backgrounds strengthen RC, highlighting the importance of cultural and linguistic congruence in improving team dynamics and healthcare delivery [6].

The empirical support for the model, in which job satisfaction precedes RC and subsequently influences intention to stay, invites a reconsideration of the traditional dynamics. Typically, RC is viewed as an antecedent of job satisfaction, which in turn influences various job outcomes. However, the findings here suggest a reverse causality, in which job satisfaction may act as a mediator to enhance relational dynamics within teams. This perspective posits that employees who are satisfied with their jobs are more likely to invest in building stronger, more effective relationships with colleagues, thereby improving RC. This improved coordination could then make employees more inclined to stay with the organization.

The testing of multiple mediation models was driven by the desire to explore the complex interplay between job satisfaction, RC, and intent to stay in the healthcare context. While this approach may seem akin to data mining, it is grounded in a rigorous exploratory analysis framework aimed at uncovering findings that may not be immediately apparent through traditional hypothesis-driven testing.

In the exploration of weak RC among administrative staff in South Tyrol, it is crucial to consider the broader implications of staff well-being on patient care. A study examining outpatient surgical clinics found that RC significantly impacts staff outcomes, such as job satisfaction, work engagement, burnout, and patient satisfaction. This underscores the dual benefit of enhancing RC, improving both staff well-being and patient experience, and sets a precedent for our investigation of the mediating role of job satisfaction in South Tyrol’s health-service context [14].

A more supportive emergency department work environment was associated with improved quality of patient care and nurses’ work outcomes, including reduced burnout, job dissatisfaction, and intention to leave [15]. In a related study, Ning et al. [16] examined the mediating roles of job satisfaction and presenteeism in the relationship between job stress and turnover intentions among primary care workers. Their findings highlight the complex interplay between job stress, personal satisfaction, and the decision to stay or leave a job. This is consistent with our findings and highlights the critical need for effective RC to reduce job stress and increase satisfaction, thereby potentially reducing turnover intention in the healthcare sector.

Given the remarkable similarities in findings from different global contexts, including Saudi Arabia [17], Australia, America, the United Kingdom, Europe [4,18,19,20,21,22,23], and East Asia [16,24,25,26,27,28], the observations on the mediating effect of RC on poor job satisfaction across all professional groups in the SABES–ASAA may not only reveal universally applicable principles but also promote a deeper understanding of how these dynamics manifest in different healthcare settings. This universal principle suggests that strategies to strengthen RC could be effectively applied in various healthcare contexts, underscoring the importance of addressing weak RC among administrative staff to improve their overall well-being and patient outcomes.

### 4.2. Relational Coordination and Willingness to Stay

The pathway from professional identity to turnover intention is mediated by job burnout, as shown among GPs in China [24]. While strong professional identity can mitigate job burnout, elevated levels of burnout significantly increase turnover intentions. This is consistent with our exploration of the influences on retention, suggesting that in addition to improving RC, addressing factors such as professional identity and burnout is critical to mitigating turnover intentions.

Research on job satisfaction among hospital doctors has identified factors such as age, gender, and specialization as significant determinants [29]. High turnover of trained GPs has been linked to excess health expenditure and burnout in GPs [30]. The cost of hospitalist turnover has been calculated to be lower than that of GPs [31] but is likely to be associated with additional costs that are more difficult to quantify. For example, high turnover can lead to low morale among the remaining team members, who have an increased workload because of staffing gaps [32]. Therefore, the high turnover of hospital doctors is also a cause of concern.

The findings on the relationship between RC, job satisfaction, and intention to resign are consistent with the identified key satisfaction and dissatisfaction factors influencing turnover among advanced practice providers in university teaching hospitals [33]. Similar to their identification of factors such as benefits, time, and collegiality within the practice, our study suggests that improving aspects of RC, such as the frequency of communication, shared goals, and mutual respect, could be critical in increasing job satisfaction and reducing turnover intention among healthcare professionals. In addition to the direct impact of RC on job satisfaction and retention, wider interpersonal dynamics within healthcare settings such as workplace ostracism play a determining role. Manninen et al. [34] demonstrated that workplace ostracism significantly reduced job satisfaction, increased stress, and worsened perceived health, with loneliness and self-esteem acting as critical mediators. As shown in another study on interpersonal dynamics in healthcare settings, ineffective collaboration significantly increased emergency nurses’ moral distress, a factor strongly associated with job dissatisfaction and potential turnover [35]. This emphasizes the importance of promoting positive workplace relationships and addressing negative behaviors, such as ostracism, to improve overall job satisfaction and retention among healthcare professionals. In primary care practices, the additional impact of structural capabilities on nurses’ job dissatisfaction and intention to leave has been documented, with both relational and structural factors playing critical roles [36]. This suggests that efforts to improve the healthcare work environment should focus not only on improving relational aspects such as communication and teamwork, but also on strengthening structural capabilities to provide comprehensive support for healthcare professionals.

Physicians experiencing burnout, with a particular impact on RC within healthcare settings, are significantly more likely to experience job dissatisfaction, regret about career choices, and increased intention to leave their jobs, highlighting a clear link between individual well-being and organizational dynamics [18]. The likelihood of patient safety incidents and diminished professionalism among burned-out physicians suggests a direct link between physician well-being and the quality of patient care [18].

Integrating the moderating role of health districts into discussions on RC, job satisfaction, and turnover intention highlights the importance of regional contexts in healthcare settings. The South Tyrol study underscores that, while RC and job satisfaction are critical in influencing staff retention, the effects of RC vary across health districts. This is consistent with global research showing that factors, such as professional identity and burnout, are mediated by local conditions, suggesting that tailored strategies are essential. Improving RC to account for local dynamics could mitigate attrition, highlighting a comprehensive approach to improve health workers’ job satisfaction and retention in diverse settings.

Improving willingness to stay despite RC and job satisfaction being low, as in the South Tyrol study, may involve several factors [37]. Strategies may include increasing organizational support, providing professional development opportunities, improving working conditions, and fostering a positive organizational culture. Recognizing and rewarding employee contributions, providing clear career paths, and ensuring fair compensation can also make a positive difference. Although low RC and job satisfaction are challenges, focusing on these areas can help mitigate their negative impacts on retention.

### 4.3. Improving Willingess to Stay: Focus on Relational Coordination and Job Satisfaction

The study findings highlight the need to implement targeted interventions to improve job satisfaction and mitigate burnout, thereby improving relationship coordination and the overall quality of care. Ensuring that the skills and expectations of medical professionals are well aligned with their job demands and that they have a harmonious relationship with their colleagues are important for improving RC and job satisfaction in healthcare settings. Xiao et al. [25] found that better alignment and harmony are correlated with increased job satisfaction and professional effectiveness among medical professionals. Implementing policies to ensure this alignment and harmony could therefore be an effective approach to address the weak RC, low job satisfaction, and low retention observed among healthcare professionals in South Tyrol.

Structured interventions in the form of nurse support programs have been shown to significantly improve job satisfaction and promote positive organizational behavior among hospital nurses [38]. These interventions included comprehensive mentoring programs, professional development workshops, and peer-support systems. Such evidence-based interventions are particularly important for addressing the challenges of low job satisfaction and retention in the healthcare sector.

The role and job satisfaction of administrative staff in the South Tyrol Health Service are relevant, considering their weak RC. Despite general job satisfaction among nursing home administrators, about 24% of administrators in a recent study reported intentions to quit [39]. By understanding the specific stressors and dissatisfiers—such as regulatory burdens, role conflicts, and staffing issues—that contribute to job dissatisfaction and turnover intentions, the South Tyrolean healthcare system should further investigate the work environment and RC of its administrators to develop targeted strategies to improve their job satisfaction and retention.

### 4.4. Limitations

It is important to acknowledge the inherent limitations of this study design and context. First, the cross-sectional nature of our survey limited our ability to establish causality between RC, job satisfaction, and retention intention. A longitudinal approach could provide a more definitive picture of these dynamics over time. In addition, reliance on self-reported measures on an ordinal scale, while convenient, may introduce bias as participants’ responses may reflect perceived ideals rather than actual experiences. Furthermore, the use of ordinally scaled variables in a linear regression, even in a moderation or mediation model, is not the best approach.

The unique bilingual and cultural landscape of South Tyrol, while a strength in understanding this specific setting, may limit the generalizability of our findings to other regions with different cultural nuances. Additionally, by focusing primarily on healthcare professionals involved in patient care and excluding additional nonclinical staff, a broader range of workplace dynamics and perspectives within the healthcare system may have been overlooked. While certain demographic and job-related factors were considered, other potential moderating variables remained unexplored and could further deepen our understanding of these relationships.

Finally, the regional specificity of our study to the SABES–ASAA means that while our findings are relevant to this context, their application to other healthcare settings requires careful consideration of local organizational cultures, healthcare policies, and workforce dynamics. Acknowledging these limitations not only grounds the study in a realistic context, but also opens up avenues for future research to build on and expand our understanding of the role of RC in healthcare professionals’ job satisfaction and retention. Future studies should also consider including control variables such as individual career aspirations, external job market conditions, and personal life circumstances, which were beyond the scope of this study but could significantly influence job satisfaction and retention outcomes.

## 5. Conclusions

This study highlights the central role of RC and job satisfaction in influencing health professionals’ willingness to stay in a culturally and linguistically diverse South Tyrol Health Service context. Job satisfaction had a significant impact on both RC and retention, with midcareer professionals having the lowest levels of satisfaction. Despite differences in job satisfaction across health districts, willingness to stay appeared to be unaffected by demographic factors, highlighting the complexity of the factors influencing health professionals’ willingness to stay. The role of RC in directly influencing employee retention through increased job satisfaction is a compelling area for further research. This suggests a reverse causality, where high RC could foster an environment that enhances job satisfaction, thereby reducing turnover intentions [40]. Further studies are needed to better understand the unique dynamics within each health district and their impact on RC, which could guide targeted interventions. Targeted interventions to enhance RC are recommended to improve job satisfaction and retention, suggesting a differentiated approach to managing healthcare workforce stability in diverse settings.

## Figures and Tables

**Figure 1 behavsci-14-00397-f001:**
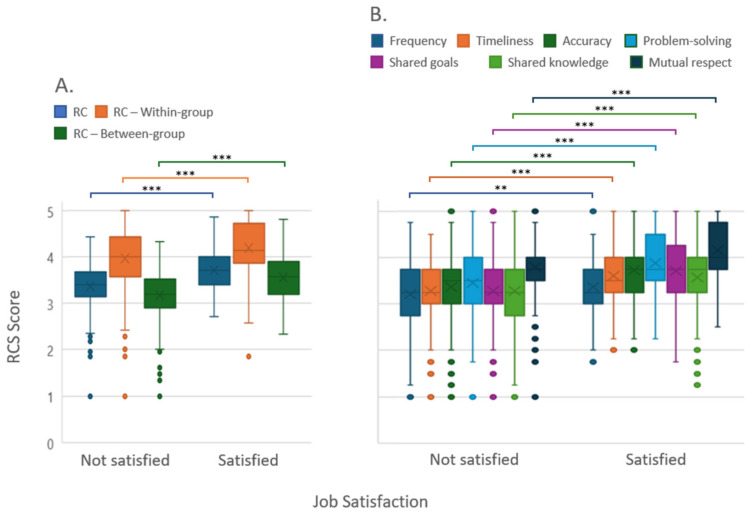
Distribution of relational coordination and its sub-dimensions for the dichotomized variable “Job Satisfaction”: (**A**) relational coordination and (**B**) subdimensions of communication and relationship. Responses indicating ‘very satisfied’ and ‘completely satisfied’ were categorized as “satisfied”, while “very dissatisfied’, ‘dissatisfied’, and ‘average satisfaction’ were categorized as “not satisfied”. RC: relational coordination; RCS: relational coordination survey. ** *p* < 0.01, *** *p* < 0.001.

**Figure 2 behavsci-14-00397-f002:**
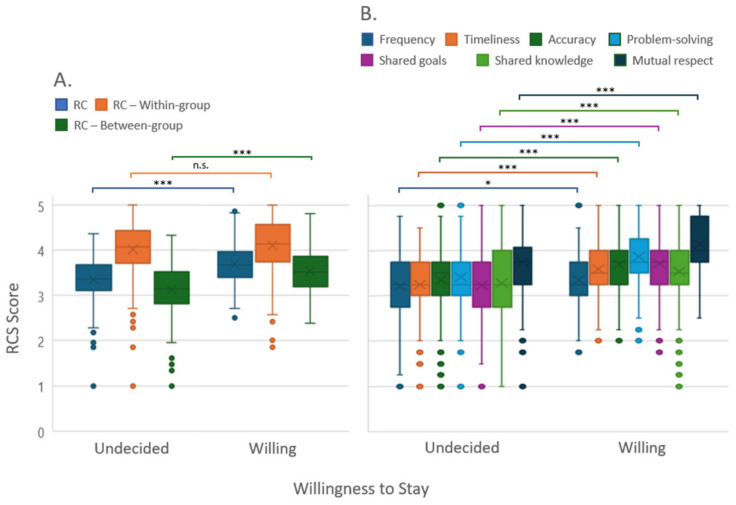
Distribution of relational coordination and its sub-dimensions for the dichotomized variable “Willingness to Stay” (box blots): (**A**) relational coordination and (**B**) subdimensions of communication and relationship. Responses of ‘completely agree’ and ‘somewhat agree’ to continue working with the current employer were classified as “willing” to stay, while ‘strongly disagree’, ‘somewhat disagree’, and ‘neither agree nor disagree’ were classified as “undecided”. RC: relational coordination; RCS: relational coordination survey. Abbreviation: n.s., not significant; * *p* < 0.05, *** *p* < 0.001.

**Figure 3 behavsci-14-00397-f003:**
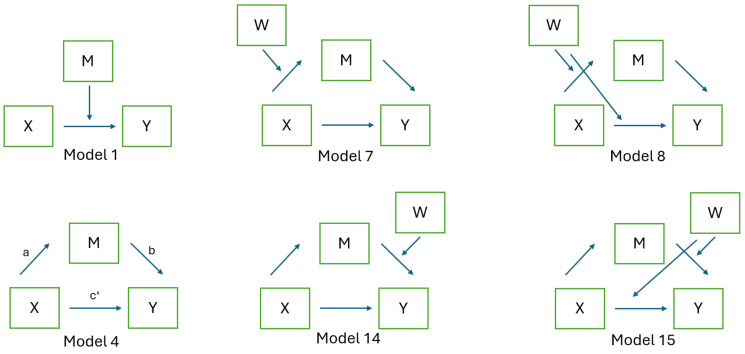
Mediation and moderation models calculated with PROCESS: Model 1 is a moderation model where ‘M’ modifies the effect of ‘X’ on ‘Y’. In Model 4, ‘M’ acts as a mediator, conveying the indirect effect of ‘X’ on ‘Y’ (a·b), alongside the direct effect (c’). Models 7, 8, 14, and 15 expand on Model 4 by incorporating ‘W’ as a moderator that affects the relationships between ‘X’, ‘M’, and ‘Y’ at different points in the path analysis. These models collectively aim to dissect the intricate relationships between variables in the context of predicting an outcome.

**Table 1 behavsci-14-00397-t001:** Dimensions of relational coordination and their impact.

Dimension	Description	Impact
Communication Frequency	Measures how frequently team members communicate	Faster problem resolution and better coordinated care
Communication Timeliness	Assesses how timely the communications occur within the team	Timely information exchange for immediate patient care needs and reducing delays
Communication Accuracy	Evaluates the accuracy of the information exchanged	Reduced errors and enhanced patient safety
Problem-solving	Looks at how teams address issues and resolve conflicts	Improves team dynamics and reduced operational friction
Shared Goals	Assesses alignment on the objectives among team members	Aligned team efforts towards common clinical and organizational outcomes
Shared Knowledge	Measures the mutual understanding of each member’s role	Enhanced role clarity, reduced overlap, and optimized resource use
Mutual Respect	Evaluates the level of respect team members have for each other	Fosters a positive work environment, increases job satisfaction, and reduces turnover rates.

**Table 2 behavsci-14-00397-t002:** Job satisfaction, and willingness to stay across professional and demographic groups cited.

Variable	Job Satisfaction				Willingness to Stay			
	Not Satisfied N (%)	Satisfied N (%)	Missing/Total	*p*-Value ^1^	Undecided to Stay N (%)	Willing to Stay N (%)	Missing/Total	*p*-Value ^1^
Total	287 (55.4)	231 (44.6)	7 (1.3)		246 (47.5)	272 (52.5)	7 (1.3)	
Professional Area				n.s.				n.s.
GPs	48 (59.3)	33 (40.7)	81		41 (50.6)	40 (49.4)		
Hospital Physicians	128 (56.4)	99 (43.6)	227		105 (46.3)	122 (53.7)		
Administrative Personnel	5 (35.7)	9 (64.3)	14		5 (35.7)	9 (64.3)		
Nurses	106 (54.1)	90 (45.9)	196		95 (48.5)	101 (51.5)		
Language				n.s.				n.s.
German	159 (53.5)	138 (46.5)	297		133 (44.8)	164 (55.2)		
Italian	128 (57.9)	93 (42.1)	221		113 (51.1)	108 (48.9)		
Gender				n.s.				n.s.
Female	184 (54.9)	151 (45.1)	335		158 (47.2)	177 (52.8)		
Male	79 (55.2)	64 (44.8)	143		66 (46.2)	77 (53.8)		
Not available	24 (60)	16 (40)	40		22 (55.0)	18 (45.0)		
Years in Service				0.019				n.s.
<10 years	57 (48.3)	61 (51.7)			47 (39.8)	71 (60.2)		
10–20 years	105 (64.4)	58 (35.6)			88 (54.0)	75 (46.0)		
>20 years	94 (53.7)	81 (46.3)			82 (46.9)	93 (53.1)		
Health District (without administrative staff)				0.024				n.s.
Health district–1	135 (61.1)	86 (38.9)	221		114 (51.6)	107 (48.4)		
Health district–2	34 (51.5)	32 (48.5)	66		28 (42.4)	38 (57.6)		
Health district–3	20 (38.5)	32 (61.5)	52		20 (38.5)	32 (61.5)		
Health district–4	81 (57.4)	60 (42.6)	141		68 (48.2)	73 (51.8)		
	Median [1Q, 3Q]				Median [1Q; 3Q]	Median [1Q; 3Q]		
Age (Average)	49 [43, 56]	48 [43; 56]		n.s.	49.5 [44; 56]	48 [43; 56]		n.s.
Years of service (Average)	17 [10, 25]	16 [7; 28]		n.s.	17.5 [10.2; 4.75]	16.0 [7.0; 27.0]		n.s.

^1^ Chi-Square test. Abbreviations: n.s., not significant; Q, quartile.

**Table 3 behavsci-14-00397-t003:** Spearman rank correlations between relational coordination and job satisfaction of willingness to stay.

Relational Coordination	Job Satisfaction (ρ/r) ^1^	*p*-Values ^2^	Willingness to Stay (ρ/r) ^1^	*p*-Values ^2^
Overall	0.384/0.415	<0.001/<0.001	0.344/0.370	<0.001/<0.001
Within-group	0.193/0.204	<0.001/<0.001	0.059/0.076	n.s./n.s.
Between-group	0.389/0.418	<0.001/<0.001	0.389/0.410	<0.001/<0.001
Dimensions				
Frequency of Communication	0.166/0.185	<0.001/<0.001	0.121/0.133	0.006/0.002
Timeliness of Communication	0.273/0.302	<0.001/<0.001	0.238/0.254	<0.001/<0.001
Accuracy of Information	0.281/0.323	<0.001/<0.001	0.245/0.288	<0.001/<0.001
Problem-solving Communication	0.344/0.359	<0.001/<0.001	0.326/0.333	<0.001/<0.001
Shared Goals	0.334/0.354	<0.001/<0.001	0.345/0.356	<0.001/<0.001
Shared Knowledge	0.261/0.252	<0.001/<0.001	0.179/0.190	<0.001/<0.001
Mutual Respect	0.323/0.345	<0.001/<0.001	0.315/0.345	<0.001/<0.001

^1^ Spearman’s rank correlation (ρ); Pearson’s correlation coefficient (r). ^2^ *p*-values for Spearman’s and Pearson’s correlation coefficients. Abbreviations: n.s., not significant.

**Table 4 behavsci-14-00397-t004:** Simple linear regression analysis with job satisfaction as dependent variable with willingness to stay and relational coordination scores as independent predictors.

Independent Predictors	Intercept	B [95% CI]	*p*-Value ^1^	R² (Model Fit)	Durbin–Watson
Willingness to stay	1.479 ***	0.518 [0.464; 0.573]	<0.001	0.406	1.914
Relational coordination					
Overall	n.s.	0.824 [0.668; 0.980]	<0.001	0.172	1.779
Within-group	1.944 ***	0.335 [0.196; 0.474]	<0.001	0.042	1.784
Between-group	0.854 ***	0.734 [0.596; 0.872]	<0.001	0.174	1.794
Dimensions					
Frequency of communication	2.281 ***	0.313 [0.169; 0.456]	<0.001	0.034	1.798
Timeliness of communication	1.719 ***	0.463 [0.337; 0.590]	<0.001	0.091	1.802
Accuracy of information	1.654 ***	0.468 [0.350; 0.587]	<0.001	0.104	1.796
Problem-solving communication	1.542 ***	0.484 [0.375; 0.593]	<0.001	0.129	1.785
Shared Goals	1.560 ***	0.502 [0.388; 0.617]	<0.001	0.125	1.812
Shared Knowledge	2.144 ***	0.341 [0.228; 0.454]	<0.001	0.063	1.792
Mutual Respect	1.340 ***	0.498 [0.381; 0.615]	<0.001	0.119	1.827

^1^ Significance level of the regression coefficient (B). Intercept: *** *p* < 0.001; n.s., not significant.

**Table 5 behavsci-14-00397-t005:** Simple linear regression analysis with intention to stay as a dependent variable with job satisfaction and RC scores as independent predictors.

Independent Predictors	Intercept	B [95% CI]	*p*-Value ^1^	R^2^ (Model Fit)	Durbin–Watson
Job satisfaction	0.933 ***	0.784 [0.702; 0.866]	<0.001	0.406	1.912
Relational Coordination					
Overall	n.s.	0.904 [0.708; 1.100]	<0.001	0.137	1.760
Within-group			n.s.	0.006	
Between-group	n.s.	0.886 [0.716; 1.057]	<0.001	0.168	1.784
Dimensions					
Frequency of communication	2.618 ***	0.276 [0.098; 0.455]	0.002	0.018	1.783
Timeliness of communication	1.887 ***	0.478 [0.320; 0.636]	<0.001	0.064	1.802
Accuracy of information	1.709 ***	0.515 [0.367; 0.662]	<0.001	0.083	1.790
Problem-solving communication	1.514 ***	0.552 [0.416; 0.667]	<0.001	0.111	1.768
Shared Goals	1.365 ***	0.621 [0.480; 0.762]	<0.001	0.127	1.830
Shared Knowledge	2.447 ***	0.316 [0.175; 0.457]	<0.001	0.036	1.771
Mutual Respect	1.107 ***	0.612 [0.468; 0.756]	<0.001	0.119	1.800

^1^ Significance level of the regression coefficient (B). Intercept: *** *p* < 0.001; n.s., not significant.

**Table 6 behavsci-14-00397-t006:** Linear regression, mediation, and moderation modeling: willingness to stay as a dependent variable.

Model (N)	Coefficient [95% CI]	*p*-Value	R^2^ (Model Fit)
Linear Regression Model (N = 518)—Predicting Willingness to Stay			0.420 ***
Relational coordination (independent predictor)	0.312 [0.135; 0.488]	0.001	
Job satisfaction (independent predictor)	0.719 [0.629; 0.808]	<0.001	
Moderation Model 1 (N = 518, Predictor: Job Satisfaction or RC)—Effects of RC, Job Satisfaction, and Interaction between RC and Job Satisfaction			0.421 ***
Effect of job satisfaction (X) on willingness to stay (Y)	0.5040 [0.226; 0.9853]	0.040	
Effect of RC (M) on willingness to stay (Y)	0.1128 [−0.3596; 0.5852]	n.s.	
Interaction between job satisfaction (X) and RC (M) on path from (X) to (Y)	0.0615 [−0.0740; 0.1970]	n.s.	
Mediation Model 4 ^1^ (N = 518, Predictor: RC)—Job Satisfaction as Mediator			0.1372 ***
Path a: RC (X) to job satisfaction (M)	0.8238 [0.6677; 0.9799]	<0.001	
Path b: Job satisfaction (M) to willingness to stay (Y)	0.7186 [0.6295; 0.8077]	<0.001	
Path c’ direct: RC (X) to willingness to stay (Y)	0.3116 [0.1348; 0.4884]	0.006	
Path a·b indirect: RC (X) to willingness to stay (Y)	0.592 [0.4684; 0.7223]	<0.001	
Path c = c’ + a·b total: RC (X) to intention to stay (Y)	0.9036 [0.7077; 1.0995]	<0.001	
Mediation Model 4 ^1^ (N = 518, Predictor: Job Satisfaction)—RC as Mediator			0.406 ***
Path a: Job satisfaction (X) to RC (M)	0.2093 [0.1697; 0.2490]	<0.001	
Path b: RC (M) to willingness to stay (Y)	0.3116 [0.1348; 0.4884]	0.006	
Path c’ (direct): Job satisfaction (X) to intention to stay (Y)	0.7186 [0.6295; 0.8077]	<0.001	
Path a·b (indirect): Job satisfaction (X) to intention to stay (Y)	0.0652 [0.0196; 0.1048]	0.011	
Path c = c’ + a·b (total): Job satisfaction (X) to intention to stay (Y)	0.7838 [0.7019; 0.8658]	<0.001	
Mediation Model 7 (N = 480, Predictor: Job Satisfaction)—RC as Mediator and Health District as Moderator in the Path Job Satisfaction to RC			0.425 ***
Path a: Job satisfaction (X) to RC (M)	0.1555 [0.0997; 0.2113]	<0.001	
Path b: RC (M) to willingness to stay (Y)	0.3362 [0.1496; 0.5228]	0.004	
Effect of health district–1, –2, or –3 (W) on RC (M)		n.s.	
Interaction of job satisfaction (X) and health district (W)		n.s.	
Interaction–1 of job satisfaction with health district–1	0.1533 [0.0140; 0.2926]	0.031	
Interaction–2 of job satisfaction with health district–2		n.s.	
Interaction–3 of job satisfaction with health district–3		n.s.	
Path c’ (direct): Job satisfaction (X) to willingness to stay (Y)	0.7106 [0.6194; 0.8018]	<0.001	
Path a·b (indirect): Interaction–1·Path b	0.0515 [0.0014; 0.1121]	BS ^2^	
Mediation Model 14 (N = 480, Predictor: Job Satisfaction)—RC as Mediator and Health District as Moderator on the Path RC to Willingness to Stay			0.446 ***
Path a: Job satisfaction (X) to RC (M)	0.2033 [0.1633; 0.2432]	<0.001	
Path b: RC (M) to willingness to stay (Y)		n.s.	
Effect of health district–1 (W1) on intention to stay (M)		n.s.	
Effect of health district–2 (W2) on intention to stay (M)	−3.9807 [−6.0527; −1.9086]	<0.001	
Effect of health district–3 (W3) on intention to stay (M)	−1.9501 [−3.3582; −0.5420]	0.007	
Interaction of RC (M) and health district (W1, W2, W3)		0.001 *	
Interaction–1 of RC with health district–1		n.s.	
Interaction–2 of RC with health district–2	1.0667 [0.5011; 1.6324]	<0.001	
Interaction–3 of RC with health district–3	0.5227 [0.1227; 0.9227]	0.010	
Path c’ (direct): Job satisfaction (X) to willingness to stay (Y),	0.7192 [0.6286; 0.8098]	<0.001	
Path (indirect): Interaction health district–2·Path a	0.2168 [0.1214; 0.3324]	BS ^2^	
Path (indirect): Interaction health district–3·Path a	0.1062 [0.0306; 0.1876]	BS ^2^	

^1^ The values shown are derived from the conditional process analysis using Model 4 of the PROCESS macro in SPSS. In the mediation and moderation models, paths are denoted by letters (a, b, c, etc.), and represent the direction and strength of the relationships between the variables. Path a: Effect of the independent variable (X) on the mediator (M). Path b: Effect of mediator (M) on dependent variable (Y). Path c’: Known as the direct effect, it represents the effect of the independent variable (X) on the dependent variable (Y) when the mediator (M) is included in the model. Path a·b: This is the indirect effect of X on Y through M, calculated by multiplying the coefficients of paths a and b. Path c: The total effect of the independent variable on the dependent variable, which is the sum of the direct (c’) and indirect (a·b) effects. ^2^ BS: Bootstrap sample; no *p*-values available. Abbreviations: RC, relational coordination; CI, confidence interval. * *p* < 0.05; *** *p* < 0.001; n.s., not significant.

## Data Availability

The data presented in this study are available upon request from the corresponding author. The data were not publicly available because the survey had politically sensitive content.

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
