# Peer review of "Impact of Relational Coordination on Job Satisfaction and Willingness to Stay: A Cross-Sectional Survey of Healthcare Professionals in South Tyrol, Italy"

_behavsci, 2024, doi:10.3390/bs14050397_

Round 1

Reviewer 1 Report

Comments and Suggestions for Authors

Abstract

The abstract provides a good case for the importance of the topic, and lays out the study questions, methods and findings quite well.  However there are two statements about findings that seem contradictory:

“The results indicated a critical influence of relational coordination on both job satisfaction and willingness to stay…”

“The findings highlight the central role of relational coordination in job satisfaction and willingness to stay and confirm that low job satisfaction increases turnover intentions, with relational coordination playing a mediating role.”

Not sure how RC can be driving two outcomes, and at the same time mediate the relationship between them. 

Introduction

Good intro overall.  There are some wording choices that I would question.

You say that : “RC is a framework that emphasizes the importance of relationships in coordinating complex tasks and is defined as a mutually reinforcing process of interaction for task integration [3].”  Rather than saying emphasizes it may be more appropriate to say “theorizes.”

Also: “Havens et al. [4] and Cramm and Nieboer [5] found that a higher RC among 47

healthcare professionals leads to better work outcomes, teamwork, and quality of patient care.” “a higher” should just be “higher"

Also in the same paragraph: “Gittell et al [6] emphasized” should be “reported”

In this sentence: “Secondly, (ii) it is hypothesized that a well-structured RC framework within healthcare teams will correlate with a stronger intention to remain in their positions, thereby potentially reducing turnover rates and strengthening the overall stability of the healthcare system” it’s not clear what you mean by “a well-structured RC framework.”  There are definitely structures that support RC, but I don’t think they are measured in your study, so you should probably just say “high levels of RC” rather than “a well-structured RC framework.”

Given that you show measures of withi- group and between-group RC below, it would be appropriate to talk about the difference between these two measures here in the introduction, from a conceptual perspective.  What does each mean?  How are they different?  And how might they impact job outcomes differently?

References to consider:

McDermott, A. M., Conway, E., Cafferkey, K., Bosak, J., & Flood, P. C. (2019). Performance management in context: formative cross-functional performance monitoring for improvement and the mediating role of relational coordination in hospitals. The International Journal of Human Resource Management, 30(3), 436-456.

House, S., Wilmoth, M., & Kitzmiller, R. (2022). Relational coordination and staff outcomes among healthcare professionals: A scoping review. Journal of Interprofessional Care, 36(6), 891-899.

Methods

Overall a good explication of your methods, including observational cross sectional research design.  But I am confused about your measure of intent to stay. 

On one hand you say: “The other item asked about their future plans with the statement, ‘I plan to leave the South Tyrolean Health Service/contract with the region as soon as possible,’ with responses on a 5-point scale ranging from a) strongly agree to e) strongly disagree.”  On the other hand you say: “For intentions to stay, responses of 'very likely' and 'somewhat likely' to continue working with the current employer were classified as 'likely to stay' (1), while 'very unlikely,’ ‘somewhat unlikely' and 'neither unlikely nor likely' were classified as 'undecided to stay' (0).” These two descriptions of the same variable are not consistent.  Please explain.

Regarding mediation, I have the same question as above in the abstract.  You say: “Mediation [15] and moderation [16] analyses were conducted to examine the potential mediating role of job satisfaction in the relationship between RC and retention, with RC as a mediator of the effect of job satisfaction on retention.”  Does job satisfaction mediate between RC and retention as you state in the first sentence?  Or does RC mediate between job satisfaction and retention as you state in the second sentence? 

Love that you also test for moderation by various demographic or job related factors.  I would suggest you include these additional questions or hypotheses in the introductory section above.

Results

The following two sets of descriptive results for job satisfaction seem to be repetitive and not completely consistent:

“The between-group and within-group RCS scores differed significantly between the groups of satisfied and unsatisfied participants. Significant differences were found for the between-group subscores in all seven dimensions and five of the seven within-group dimensions. Only the frequency and timeliness of communication within groups did not show significant differences between satisfied and unsatisfied participants.”

“Between the two categories of the dichotomized variable job satisfaction, 'satisfied’ and 'not satisfied,' highly significant differences were found for RC, within-group RC, and between-group RC, as well as for the RC dimensions 'accuracy of information,' 'timeliness of information,' 'problem-solving communication,' 'shared goals,' 'shared knowledge' and 'mutual respect.' Only the dimension 'frequency of information' produced fewer but still significant results (Figure 1).”

Please choose one or the other of the above reports. 

When you report intent to stay descriptive results, there are other issues:

“For the dichotomized variable willingness to stay, 'unlikely to stay' and 'undecided to stay,' RC and external RC showed a highly significant difference between the two categories, while no significant difference was found for internal RC (Figure 2). Even all RC dimensions are 'accuracy of information,' timeliness of information,' 'problem-solving communication,' 'shared goals,' 'shared knowledge,' ' and 'mutual respect.' Only the dimension 'frequency of information' produced fewer but still significant differences.”

First of all “unlikely to stay” and “undecided to stay” don’t make sense for the dichotomized variable and don’t reflect what you wrote above.  Secondly, internal and external RC should probably be labeled the same as you have above – within group and between group RC.

I see that you tested multiple mediation models involved job satisfaction, RC and intent to stay, including the two different versions that show up in the abstract and the methods section.  The model that gets the best empirical support is the one that is not supported by existing theory:

Job Sat >>> Relational Coordination >>> Intent to Stay

I would love to hear your hypothesis for this model.  I can’t think of one!

One reaction to all the many models you test is: it does not feel like a theory driven process, but rather like a data mining process in which you test every possible combination of variables ot see how they are related.  Is that what you intended to do?  Is this an acceptable way to do empirical work?  It may be – I just don’t know!

Discussion

This interpretation of your findings is very interesting: “The findings indicate that health districts do not directly affect the relationship between job satisfaction and retention. Instead, their influence manifests indirectly through RC, highlighting a nuanced interaction effect where the RC's impact on retention varies by health district. This suggests that the role of job satisfaction in retaining healthcare professionals is complemented by the specific dynamics of RC within different health districts, adding a layer of complexity to the understanding of retention factors in the South Tyrol healthcare system.”  I’m not sure I understand however!

You also say: “The findings of the South Tyrol study align with international research, illustrating the critical role of RC as a mediator in health care settings.” However RC theory and findings suggest that RC is a mediator between organizational structures and outcomes, where cross-cutting structures help to support RC and siloed structures serve to weaken RC, thus impacting outcomes.

RC to my knowledge has never been studied as a mediator between job satisfaction and turnover.  It is certainly possible but you should provide some interpretation of this finding, given it was not hypothesized in your introduction.

Discussion

The discussion is very confusing as it is currently written.  The causal relationships you are talking about are not clear.  I think once you clarify what you have found, and give us some ideas about what it means, it will be easier to revise your discussion.

Comments on the Quality of English Language

Some need for editing.

Reviewer 2 Report

Comments and Suggestions for Authors

Authors

Nice work. I would have liked to have seen a description of RC included as a table for those not familiar with the theoretical framework as well as a description of the health districts - similarities and difference. For example, does the cross-cultural make-up of each differ drastically? Have different staffing levels? Different skill sets of key personnel?

The discussion/conclusion section of the paper needs more depth, especially around the various health districts and how their inherent differences might account for some of the findings.  Is further study needed? Qualitative study perhaps? Interventional and if so, what might the interventions be? 

Reviewer 3 Report

Comments and Suggestions for Authors

This study examines how relational coordination, impacts outcomes among healthcare professionals in Italy. The paper can also explore the negative aspects of healthcare work in the intro, you can check the work: Violence against health care workers in China, 2013–2016: evidence from the national judgment documents. Human Resources for Health, 

The paper does not specify the response rate of the survey, which is critical in assessing the representativeness of the sample.

The study focuses primarily on relational coordination, job satisfaction, and willingness to stay, but control variables individual career aspirations, external job market conditions, and personal life circumstances might also influence these outcomes.

The  should emphisize contribution by offering more specific recommendations for healthcare administrators and policy makers.The paper sometimes repeats information in various sections so streamline content and avoid redundancy such as: If the role of relational coordination in enhancing job satisfaction is detailed in the introduction, avoid extensive repetition of the same points in the discussion unless adding new information or insights.

There is a scarce literature review done, and only in the introduction section. More background on health workers can be explored, such as in the recent studies on nurses mental health: Mediating effect of coping styles on the association between psychological capital and psychological distress among Chinese nurses: a cross-sectional study. Journal of Psychiatric and Mental Health Nursing, 

How nursing students’ risk perception affected their professional commitment during the COVID-19 pandemic: the mediating effects of negative emotions and moderating effects of psychological capital. 

This cross-sectional survey included general practitioners, hospital physicians, nurses which is a heteregeneous sample, why didnt you collect only from one type of healthcare staff?

Address all limitations.

Comments on the Quality of English Language

proofread the paper further

Round 2

Reviewer 1 Report

Comments and Suggestions for Authors

Thank you for the revisions.

Reviewer 3 Report

Comments and Suggestions for Authors

none 

Comments on the Quality of English Language

none